# Alterations of in vivo CA1 network activity in Dp(16)1Yey Down syndrome model mice

Matthieu Raveau[1], Denis Polygalov[2], Roman Boehringer[2], Kenji Amano[1], Kazuhiro Yamakawa[1]*, Thomas J McHugh[2]*

[1]Laboratory for Neurogenetics, RIKEN, Brain Science Institute, Saitama, Japan; [2]Laboratory for Circuit and Behavioral Physiology, RIKEN, Brain Science Institute, Saitama, Japan

**Abstract** Down syndrome, the leading genetic cause of intellectual disability, results from an extra-copy of chromosome 21. Mice engineered to model this aneuploidy exhibit Down syndrome-like memory deficits in spatial and contextual tasks. While abnormal neuronal function has been identified in these models, most studies have relied on *in vitro* measures. Here, using *in vivo* recording in *the* Dp(16)1Yey model, we find alterations in the organization of spiking of hippocampal CA1 pyramidal neurons, including deficits in the generation of complex spikes. These changes lead to poorer spatial coding during exploration and less coordinated activity during sharp-wave ripples, events involved in memory consolidation. Further, the density of CA1 inhibitory neurons expressing neuropeptide Y, a population key for the generation of pyramidal cell bursts, were significantly increased in Dp(16)1Yey mice. Our data refine the 'over-suppression' theory of Down syndrome pathophysiology and suggest specific neuronal subtypes involved in hippocampal dysfunction in these model mice.

DOI: https://doi.org/10.7554/eLife.31543.001

*For correspondence:
yamakawa@brain.riken.jp (KY);
tjmchugh@brain.riken.jp (TJMH)

**Competing interests:** The authors declare that no competing interests exist.

## Introduction

Down syndrome (DS), with an incidence of one in 700 to 1000 live births, is the most common genetic cause of intellectual disability (*Parker et al., 2010*). It results from a triplication of chromosome 21 (HSA21) and leads to a wide spectrum of phenotypes, among which craniofacial malformation and intellectual deficits are the most striking for their full penetrance (*Antonarakis et al., 2004*). The behavioral phenotypes in DS include learning and memory deficits affecting both verbal and non-verbal cognition (*Chapman and Hesketh, 2000*) and poor performance in spatial memory tasks (*Lavenex et al., 2015*). Cellular defects have been observed in post-mortem brains of DS fetuses (*Contestabile et al., 2007*; *Guidi et al., 2008*) but the mechanisms affecting function in the developing and adult DS brain have remained elusive. Accumulating data, however, support the idea that a shift in the balance between inhibitory and excitatory transmission may contribute to DS learning phenotypes, particularly in the hippocampus (*Kleschevnikov et al., 2012*; *Kleschevnikov et al., 2004*). This has led to the proposal that targeting alterations in GABAergic signaling may be a viable therapeutic target (*Contestabile et al., 2017*), however evidence for this on the circuit level is scarce.

A large portion of HSA21 possesses a syntenic region on mouse chromosome 16 (MMU16), conserving many of the genes in the same order and relative orientation. Mouse models of DS have thus been developed to circumvent the limited access to brain structures and the genetic variance in the DS population. These models, carrying various lengths of chromosomal duplications, recapitulate most of the DS-like characteristics (*Lana-Elola et al., 2011*; *Dierssen, 2012*). In particular, models

carrying some of the largest duplications, Ts65Dn and Dp(16)1Yey, display learning and memory deficits associated with a loss of hippocampal long-term potentiation (*Reeves et al., 1995*; *Kleschevnikov et al., 2004*; *Yu et al., 2010a*). Similar phenotypes have been described in the trans-chromosomic Tc1 model carrying a nearly complete HSA21 chromosome (*O'Doherty et al., 2005*; *Morice et al., 2008*). These models thus mimic the human condition in regards to hippocampal dependent spatial and working memory deficits.

In rodents the hippocampus is crucial for the encoding and consolidation of spatial memory (*Jarrard, 1993*). In the CA1 region pyramidal neurons exhibit spatial receptive fields, referred to as 'place fields', which manifest as discrete locations in the environment where a given cell's firing rate increases (*O'Keefe and Dostrovsky, 1971*). Further, during movement the hippocampus exhibits rhythmic population activity in the theta-band (6–12 Hz) which contributes to memory encoding by temporally organizing pyramidal cell spiking (*Buzsáki, 2002*; *O'Keefe and Recce, 1993*). Following exploration, when animals are resting or sleeping, the hippocampal local field potential is dominated by short high-frequency oscillations (~100 ms, 80–250 Hz) referred to as sharp wave ripples (SWRs; *Buzsáki, 2015*), events shown to be critical for memory consolidation (*Ramadan et al., 2009*; *Ego-Stengel and Wilson, 2010*). During SWRs CA1 pyramidal cells display temporally precise spiking that replays activity related to past behavioral episodes (*Karlsson and Frank, 2009*; *Girardeau et al., 2009*). Thus both during encoding and consolidation the precise timing of hippocampal spiking is a crucial feature of creating a functional memory trace (*Buzsáki, 1989*). This precision is under the tight control of a diverse family of inhibitory interneurons which regulate the generation of bursts in pyramidal cells, their spike timing within theta during exploratory behavior (*Royer et al., 2012*) and the synchronization required to generate and maintain ripple events (*Pangalos et al., 2013*; *Cutsuridis and Taxidis, 2013*). The tight inhibitory system grafts onto the excitatory input, involving direct input to the CA1 from the entorhinal cortex as well as excitation arriving via the tri-synaptic circuit, in which information flows through the dentate gyrus to the CA3 and/or CA2 and then on to CA1 (*Jones and McHugh, 2011*), and ensures proper activation and synchronization of the hippocampal network.

*In vitro* approaches have identified synaptic plasticity deficits in the hippocampus of DS mice (*Belichenko et al., 2015*). Though long-term potentiation deficits in the dentate gyrus (*Morice et al., 2008*) and mild instability of CA1 place fields (*Witton et al., 2015*) have been observed *in vivo* in Tc1 mice, no extensive electrophysiological characterization of a DS mouse model in freely moving conditions has been reported. Here we use the Dp(16)1Yey model carrying a large and stable tandem duplication on the MMU16 (*Figure 1—figure supplement 1*) to circumvent the issue of mosaicism seen in Tc1 mouse brain (~66% chromosome retention in the brain; *O'Doherty et al., 2005*). Using single-unit activity and LFP recording in the dorsal CA1 of freely moving Dp(16)1Yey and control mice both during exploratory behavior on a linear track and periods of rest we analyzed CA1 physiology at the single cell and population levels. We combined this electrophysiological study with an immunohistochemical approach to help elucidate possible circuit mechanisms affecting CA1 function in DS mice.

## Results

### Decreased bursting and complex spiking and abnormal spatial encoding during exploration in the CA1 of Dp(16)1Yey mice

In order to investigate hippocampal spatial coding Dp(16)1Yey (Dp16; N = 6) and wild-type littermate control (WT; N = 5) mice were implanted with recording electrodes in the dorsal CA1 pyramidal cell layer and allowed to explore a linear track (*Figure 1A*). We observed no difference between the groups in average velocity (Dp16: 4.17 ± 0.60 cm/s; WT: 4.57 ± 0.19 cm/s; p=0.542) or total distance traveled (Dp16: 4857.92 ± 424.04 cm; WT: 5695.24 ± 318.18 cm; p=0.128). We first compared local field potentials (LFP) recorded in CA1 while the mice were exploring and found no difference between the groups across all frequencies, including the theta (6–12 Hz) and gamma bands (30–100 Hz) prominent during exploratory behavior (*Figure 1—figure supplement 2A*). We then isolated and sorted single-unit activity from CA1 pyramidal cells and examined their spatially modulated firing (WT: n = 256; Dp16: n = 259). Place fields in Dp16 mice were more diffuse than those in their control littermates (*Figure 1B*). Quantitative analysis confirmed that in the DS mice the place fields

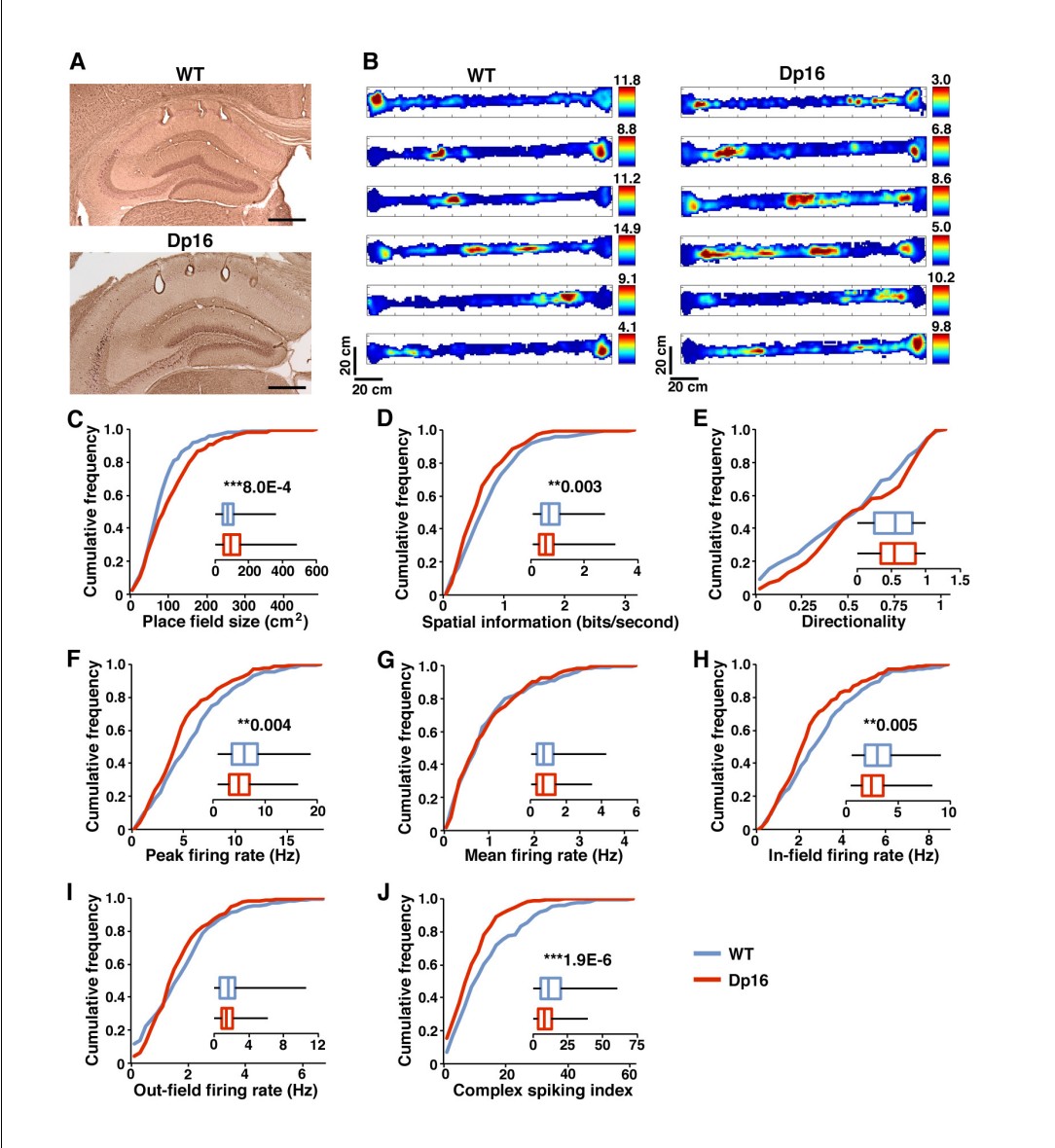

**Figure 1.** Poorer spatial coding and decreased bursting in Dp(16)1Yey mice during exploration. (**A**) Example of tetrode positioning in the dorsal CA1 pyramidal cell layer of WT (upper) and Dp(16)1Yey (lower) mice. (**B**) Examples of color-coded firing rate maps of CA1 place cells during exploration of a 170 × 10 cm linear track. Red indicates peak firing rate in Hz (value for each cell indicated on the right of each map) while blue represents no firing. (**C**) Place field size was significantly larger in Dp(16)1Yey mice. (**D**) The spatial information content encoded per second by pyramidal cells was significantly lower in Dp(16)1Yey mice. (**E**) The firing rate directionality was comparable between the Dp(16)1Yey and WT groups. Pyramidal cells' peak firing rate (**F**) was significantly lower in the Dp(16)1Yey group whereas their mean firing rate (**G**) remained comparable to the WT group. The firing rate within the place field (**H**) was significantly lower in Dp(16)1Yey mice, but this was not the case outside the place fields (**I**). The complex spiking index (**J**) was significantly decreased in the Dp(16)1Yey group. Statistical significance was assessed using Mann-Whitney U-test with significance set at (**) $p < 0.01$ and (***) $p < 0.001$. Scale bars in (**A**) correspond to 500 μm.

DOI: https://doi.org/10.7554/eLife.31543.002

The following source data and figure supplements are available for figure 1:

**Source data 1.** Pyramidal cells characteristics during exploratory activity - full data set.

DOI: https://doi.org/10.7554/eLife.31543.005

**Figure supplement 1.** Human – mouse synteny allows modeling Down syndrome in mice.

DOI: https://doi.org/10.7554/eLife.31543.003

**Figure supplement 2.** LFP power and phase locking properties of CA1 place cells are conserved in Dp(16)1Yey mice.

DOI: https://doi.org/10.7554/eLife.31543.004

were significantly larger (*Figure 1C*) and sparser (Firing rate map sparsity: WT: 0.319 ± 0.010; Dp(16) 1Yey: 0.359 ± 0.011; p=0.0189), resulting in a significant decrease in the spatial information encoded by these neurons (*Figure 1D*). On a linear track hippocampal place cells exhibit directionality, a bias to fire more in one direction than the other. The 'directionality index' of pyramidal cells in Dp(16) 1Yey samples was comparable to that of their WT littermates suggesting that this aspect of coding was conserved (*Figure 1E*).

CA1 pyramidal cells from Dp16 animals had a significantly lower peak firing rate during exploration (*Figure 1F*) but an average firing rate comparable to control littermates (*Figure 1G*). On a finer level we observed that Dp16 place cells displayed a significantly lower in-field firing rate but fired normally outside of their place field (*Figure 1H,I*). Taken together, these results suggest that place cells in the DS mice may lack the ability to generate the normal burst patterns of firing seen when the animal navigates through the receptive field (*Ranck, 1973*; *Quirk and Wilson, 1999*). To investigate if the decrease in peak firing rate we observed reflected a loss of the ability to produce the complex spike bursts typical of pyramidal cells we compared the complex spike index (CSI; *McHugh et al., 1996*) between the groups and found it to be drastically lower in the Dp16 mice (*Figure 1J*).

During movement the spiking of pyramidal cells is strongly modulated by the underlying theta oscillation (*Siapas et al., 2005*). We next asked if the changes in bursting seen in Dp16 mice impacted this aspect of spike timing. The proportion of significantly theta-modulated cells was not significantly different between Dp16 and control mice (WT: 121/256, 47.3%; Dp16: 115/259, 44.4%, Chi-square test p=0.514) and the firing probability of significantly theta-modulated CA1 place cells as a function of theta phase showed a highly similar pattern between the groups (*Figure 1—figure supplement 2B*). The preferred phase of theta (*Figure 1—figure supplement 2C*) and the concentration of the firing probability (*Figure 1—figure supplement 2D*; *Siapas et al., 2005*) were also similar between the groups. Further quantification of the LFP/single unit activity interaction revealed a similar degree of phase divergence (Kullback-Leibler divergence, *Figure 1—figure supplement 2E*) and an equivalent modulation index (*Figure 1—figure supplement 2F*; *Gu et al., 2013*). Overall, these results indicate that theta-modulation was unaltered in the Dp16 mice.

## Alterations in the ensemble code for space in Dp(16)1Yey mice

The hippocampal code of space is best reflected in the coordinated activity across the population of neurons (*Wilson and McNaughton, 1993*). Thus, we next investigated the spatial representation at the ensemble level. For that purpose, we generated population vectors for all place cells recorded and computed autocorrelation matrices for control (*Figure 2A*) and Dp16 mice (*Figure 2B*). This strategy provides a visualization of the fine tuning of pyramidal cell activity for the representation of space at the population level (*Resnik et al., 2012*). Given the directional preference of single place cells, we separated left and right laps on the track and examined their auto and cross-correlations independently (*Figure 2A and B*). In the resulting matrices quadrants II and III correspond to the correlation of a population vector (quadrant II: left laps, quadrant III: right laps) with itself and hence, points along the diagonal reach maximal values, while points away from the diagonal correspond to the correlation between population activity at different positions, with increasing distance from the diagonal reflecting increasingly distant positions on the track. In both the control (*Figure 2A*) and Dp16 mice (*Figure 2B*) the correlation decreased rapidly as a function of distance from the diagonal. The averaged values of the correlation between population vectors as a function of distance confirmed this pattern, with no significant differences between the groups, except for largely distant population vectors for which the correlation was lower in the Dp16 animals (*Figure 2C*).

To assess absolute spatial coding, independent of the direction of movement, we compared the similarity of the population vectors for laps in opposite directions in the linear track (quadrants I and IV). Interestingly, whereas there was a clear diagonal band of structure obvious in the control data (*Figure 2A*), this was not present in the Dp16 data (*Figure 2B*). In control mice the averaged values of the correlation between population vectors as a function of distance showed a significant central peak, with the maximal value slightly shifted to the left (*Figure 2D*). These findings are similar to previous reports and this shift has been suggested to reflect prospective coding of future locations (*Resnik et al., 2012*). The Dp16 population vectors however failed to show this central peak and correlation values remained significantly lower than in WT mice, even for similar positions on the track. Taken together, these data suggest that while the direction dependent representation of position is

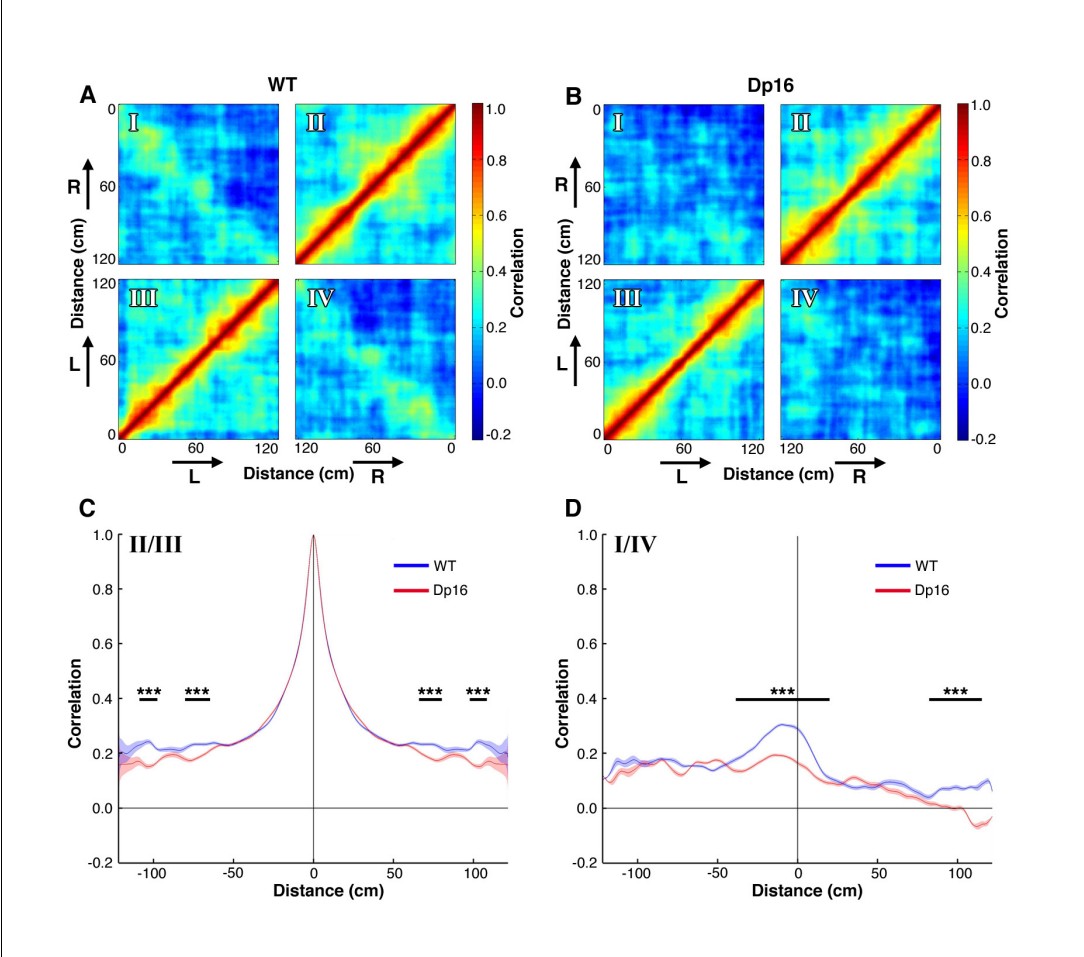

**Figure 2.** Population level coding of position is altered in Dp(16)1Yey mice. (**A**) Spatial autocorrelation matrix of the population vectors in WT mice. Lower left (III) and upper right (II) quadrants represent the correlation of population vectors encoding movements in the same direction. Upper left (I) and lower right (IV) quadrants represent the correlation for opposite directions. Arrows indicate the movement direction (L: left laps; R: right laps). (**B**) Spatial autocorrelation matrix of the population vectors in Dp(16)1Yey mice. (**C**) Average correlation as a function of track distance for population vectors encoding movement in the same direction. The blue and red lines represent average values of the quadrants II and III of the cross-correlation matrices displayed on panels A and B respectively. The shaded areas correspond to the standard error of mean (S.E.M) of these values. The dominant central peak indicates a strong positional encoding independent from the location of the place fields in the track, in both WT and Dp(16)1Yey mice. (**D**) Same as in (**C**) for population vectors encoding movements in opposite direction, averaged across quadrants I and IV of the cross-correlation matrices shown on panels A and B.. The peak in the center indicates a mild correlation in the encoding of the animal's position between movements in opposite directions in the WT group. Dp(16)1Yey mice however failed to show a similar pattern. Statistical significance was assessed using Wilcoxon ranksum test with significance set at (***) p<0.001.

DOI: https://doi.org/10.7554/eLife.31543.006

not significantly altered in Dp(16)1Yey mice, the direction independent coding of position observed in control mice is greatly decreased.

## Decreased bursting during memory consolidation in Dp(16)1Yey mice

Following exploration of the linear track mice were placed in a familiar rest box for 30 min and CA1 pyramidal cell activity was recorded to investigate activity during the post-run phase. During this quiescent period the peak and average firing rate of CA1 pyramidal cells were not significantly different between Dp16 and WT mice (*Figure 3A,B*). However, similar to what we observed during exploration, the complex spiking index was significantly lower in the DS model mice, suggesting that the deficit in the generation of complex spikes could impact off-line memory consolidation (*Figure 3C*). Plotting the inter-spike interval (ISI) distribution of all spike trains from WT pyramidal cells revealed a

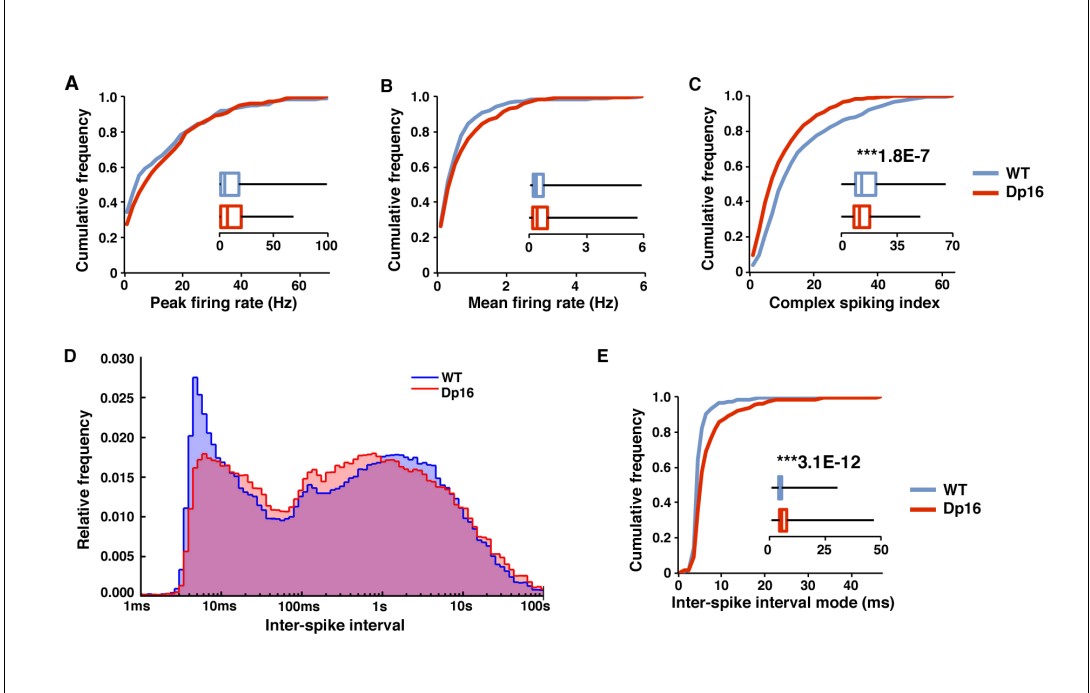

**Figure 3.** Decreased bursting of CA1 pyramidal cells in Dp(16)1Yey mice during post-exploratory rest. During post-exploratory quiescent periods the peak firing rate (A) and mean firing rate (B) were not significantly different between the WT and Dp(16)1Yey groups. (C) The complex spiking index was significantly decreased in Dp(16)1Yey CA1 pyramidal cells. (D) The ISI distribution shows two populations of events in CA1 pyramidal cells from WT mice. The peak of events with an ISI between 2 and 15 ms correspond to spikes included in complex bursts. The second large population of events with higher ISI represents single isolated spikes. In the Dp(16)1Yey the peak of short ISI events was smaller than in WT, whereas the population of isolated single spikes tended to be larger. (E) The inter-spike interval (ISI) mode, defined as the most probable inter-spike interval for each recorded neuron, was significantly longer in the Dp(16)1Yey group than in their WT littermates. Statistical significance was assessed using Mann-Whitney U-test with significance set at (***) p<0.001.

DOI: https://doi.org/10.7554/eLife.31543.007
The following source data is available for figure 3:

**Source data 1.** Pyramidal cells characteristics during awake-rest - full data set.
DOI: https://doi.org/10.7554/eLife.31543.008

clear narrow peak at short ISIs (~2 to 15 ms) and a second wide peak at longer ISIs (≥100 ms; *Figure 3D*). These populations correspond to ISIs resulting from bursts (including complex spikes) and those representing the time between bursts and/or single spikes respectively. In contrast, the ISI distribution of spike trains recorded from Dp16 pyramidal cells showed a reduction of the short ISI peak and a slight increase at longer ISIs (*Figure 3D*). Accordingly, the ISI mode, defined as the most frequent ISI for a given cell, was significantly longer in the Dp16 data compared to controls (*Figure 3E*). An increase in the ISI mode, which preferentially reflects the frequent short ISIs generated by bursts, without a significant change in the mean firing rate suggested that spikes generated by CA1 pyramidal cells in Dp16 mice were sparser and less likely to be involved in bursts. Overall, these data indicate that in Dp(16)1Yey mice the ability of pyramidal cells to fire bursts of spikes is impaired.

For a more detailed quantification we isolated bursts of spikes in both groups and examined their properties (*Table 1*). The number of bursts per minute was significantly decreased in the Dp16 group, whereas the mean inter-burst interval was significantly increased, both during exploration and rest. Average burst duration was significantly longer in DS mice during exploration, but this was not the case during rest. Moreover, the percentage of spikes participating in bursts, as well as the average number of spikes per burst, were significantly lower in Dp16 place cells, both during memory encoding and consolidation. These findings indicate that the decrease in complex spiking index

**Table 1.** Bursting characteristics of CA1 place cells during exploration and rest.

| | Linear track | | | | | | | Rest | | | | | | |
|---|---|---|---|---|---|---|---|---|---|---|---|---|---|---|
| | WT | | | Dp16 | | | | *p-value* | WT | | | Dp16 | | | | *p-value* |
| Bursts per min | 7.901 | ± | 0.532 | 5.777 | ± | 0.383 | *** | *0.0007* | 7.152 | ± | 0.685 | 5.741 | ± | 0.468 | * | *0.0315* |
| Mean Inter-burst Interval (s) | 18.940 | ± | 1.692 | 25.145 | ± | 1.907 | *** | *0.0002* | 30.061 | ± | 2.141 | 56.144 | ± | 6.640 | * | *0.0320* |
| Mean burst duration (ms) | 7.720 | ± | 0.186 | 8.435 | ± | 0.318 | *** | *2.67E-07* | 12.718 | ± | 4.251 | 8.281 | ± | 0.072 | | 0.1305 |
| % of spikes in burst | 33.1% | ± | 0.8% | 24.4% | ± | 0.7% | *** | *2.17E-13* | 29.7% | ± | 0.8% | 22.3% | ± | 0.7% | *** | *9.58E-12* |
| Nb. Spikes per burst | 2.264 | ± | 0.008 | 2.243 | ± | 0.009 | * | *0.0110* | 2.374 | ± | 0.010 | 2.264 | ± | 0.009 | *** | *1.59E-14* |

DOI: https://doi.org/10.7554/eLife.31543.009
The following source data available for Table 1:
Source data 1. CA1 bursting analysis - full data set.
DOI: https://doi.org/10.7554/eLife.31543.010

observed is likely explained by an overall loss of firing during bursts on the single cell level, impacting both the encoding and consolidation of spatial information.

## Sharp wave ripples properties are affected and place cells fire less during ripples in Dp(16)1Yey CA1

The decrease in bursting during rest in Dp16 mice (*Figure 3C*) suggests possible alterations in temporal coding, so we next analyzed LFP traces during the post-exploratory rest period to assess if population synchronicity was also affected. During sleep and quiet wakefulness the CA1 LFP is defined by frequent SWR events, fast oscillations of about 100 ms that reflect the synchronous activity of large population of pyramidal cells (*Buzsáki, 2015*; *Cutsuridis and Taxidis, 2013*). We detected ripple events in the LFP and compared their characteristics across the groups (*Figure 4A*). The number of ripples per minute and mean inter-ripple intervals were not significantly different between Dp16 and WT mice (*Figure 4B,C*). Although there was a slight shift in the population distribution towards lower intrinsic frequencies in the DS mice (*Figure 4—figure supplement 1*), we found no significant difference in the peak ripple frequency (*Figure 4D*). However ripples in the Dp16 mice were significantly smaller in amplitude (*Figure 4E*) and shorter in duration than those found in controls (*Figure 4F*).

Given the changes in pyramidal neuron bursting and SWR amplitude and duration we asked if this altered the firing of single cells during SWRs. The average number of spikes per cell per ripple was slightly, yet significantly, lower in Dp16 mice (*Figure 4G*) and pyramidal cells participated in a significantly lower proportion of ripple events (*Figure 4H*). Further, plotting the populations of cells according to their ripple participation rates illustrated a significant shift toward lower values in the DS model, with a peak at 5–10% in the Dp16 mice compared to a peak at 10–15% in the WT group (*Figure 4I*). These changes in single unit activity were unrelated to the decrease in ripple amplitude in the Dp16 mice, as we found significant decreases in ripple-related spiking and event participation across a wide range of ripple detection thresholds (*Figure 4—figure supplement 2A,B*). To further confirm that these changes were not attributable simple to the smaller amplitude of ripples in the Dp16 group we also detected the onset and offset of the events using only multiunit spiking activity and again observed a significant decrease in ripple amplitude, duration, number of spikes and participation in ripple events (*Figure 4—figure supplement 3*).

## Density of neuropeptide Y expressing interneurons is increased in the CA1 of Dp(16)1Yey mice

The complex spiking of pyramidal cells and their synchronized activity during population events are under tight control of local inhibitory circuits within CA1 (*Milstein et al., 2015*; *Royer et al., 2012*; *Cutsuridis and Taxidis, 2013*). To address if alterations in this complex network in Dp(16)1Yey mice could underlie the changes in *in vivo* physiology we observed we used immunohistochemistry to quantify the main populations of interneurons in the different layers of the CA1, employing the

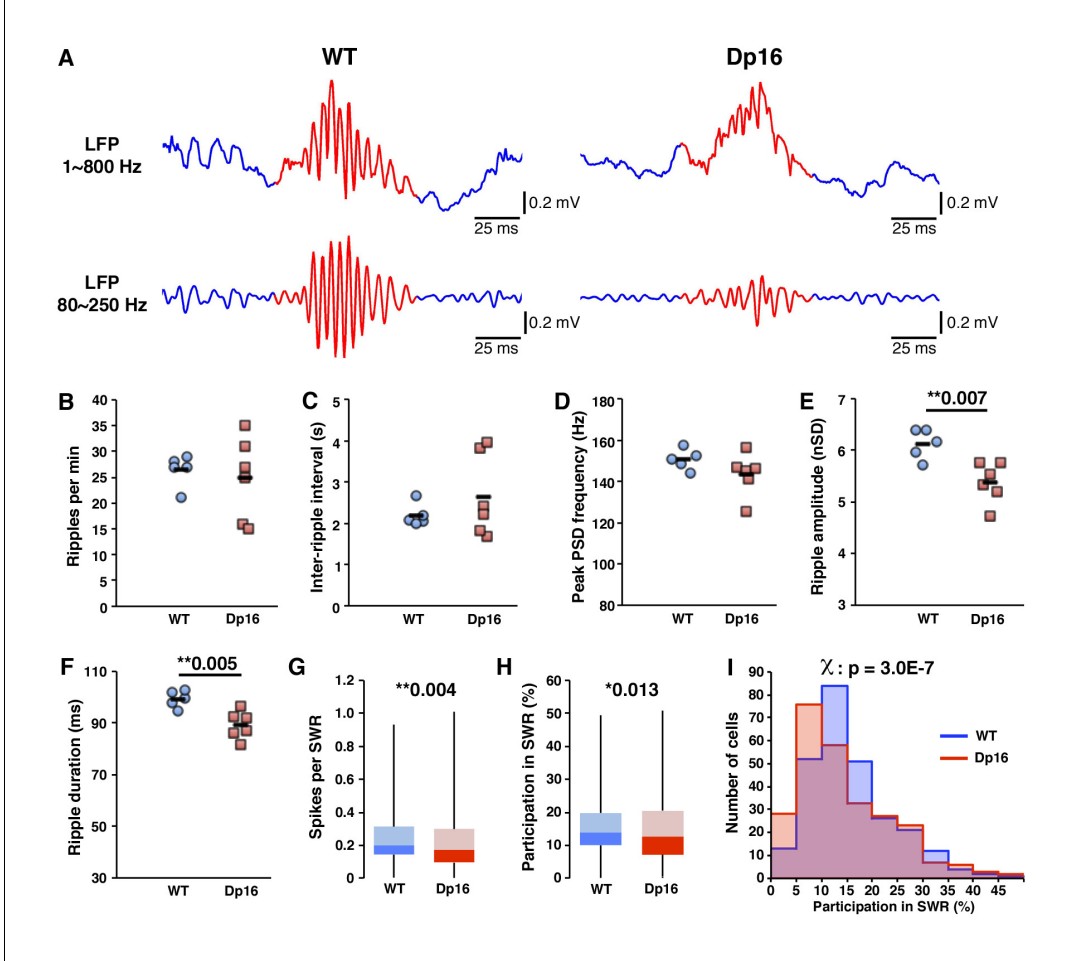

**Figure 4.** Decreased activity of CA1 pyramidal cells during ripple events in Dp(16)1Yey mice. (A) Examples of local field potentials (LFP), non-filtered (upper traces) or filtered for the ripple frequency range (lower traces). The occurrence of ripple events (B), the inter-ripple interval (C) and the peak ripple frequency (D) were not significantly different between Dp(16)1Yey and WT mice. Ripple amplitude (E) and duration (F) were significantly decreased in the LFPs of Dp(16)1Yey mice. The number of spikes per cell per ripple (G) and the participation of individual cells in ripple events (H) were significantly lower in CA1 pyramidal cells of Dp16 mice. (I) The proportion of cells plotted as a function of their participation rate in sharp-wave ripples showed a significant shift toward lower participation rates, with Dp(16)1Yey pyramidal cells peaking at 5–10% participation whereas the peak was at 10–15% in the WT group. Each marker in (B) thru (F) represents average value from all events in an individual mouse. The origin of the Y-axis in D, E and F was set at the threshold used for ripple detection. Statistical significance was assessed using one-way ANOVA (B–F) or Mann-Whitney U-test (G, H) with significance set at (*) $p<0.05$ and (**) $p<0.01$.

DOI: https://doi.org/10.7554/eLife.31543.011

The following source data and figure supplements are available for figure 4:

**Source data 1.** Sharp wave ripples analysis - full data set.
DOI: https://doi.org/10.7554/eLife.31543.015

**Figure supplement 1.** Sharp wave ripple peak frequency was not significantly affected in Dp(16)1Yey mice.
DOI: https://doi.org/10.7554/eLife.31543.012

**Figure supplement 2.** Decreased activity of CA1 pyramidal cells during ripple events in Dp(16)1Yey mice is not related to ripple amplitude.
DOI: https://doi.org/10.7554/eLife.31543.013

**Figure supplement 3.** Decreased activity of CA1 pyramidal cells during ripple events in Dp(16)1Yey mice is not related to ripple detection bias.
DOI: https://doi.org/10.7554/eLife.31543.014

molecular markers parvalbumin (PV), somatostatin (SST) and neuropeptide Y (NPY) to differentiate among classes of interneurons.

Populations of NPY positive interneurons (*Figure 5A*) were observed in the *stratum oriens* (SO), *stratum pyramidale* (SP) and *stratum lacunosum moleculare* (SLM). The density of these NPY positive

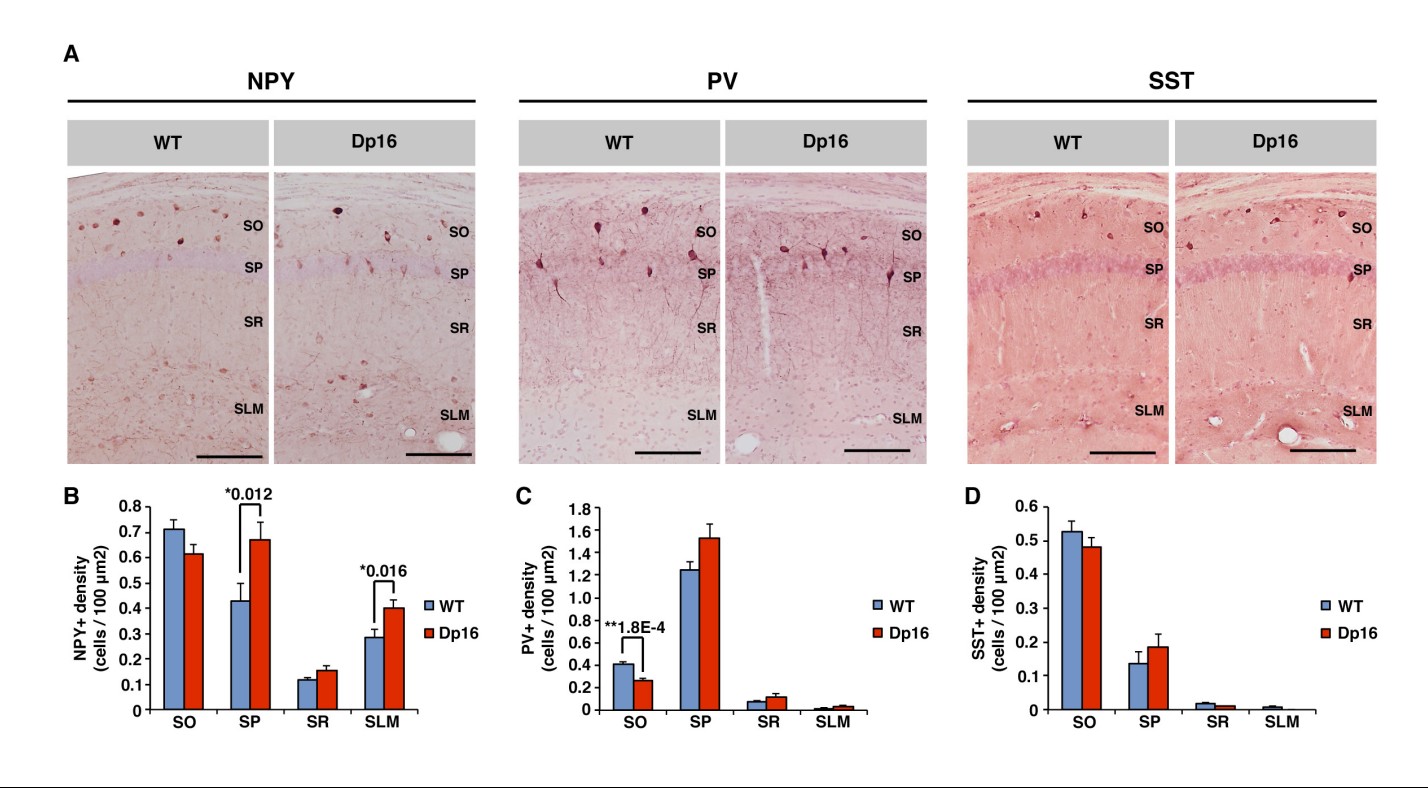

**Figure 5.** Neuropeptide Y-positive interneuron density is increased in the CA1 of Dp(16)1Yey mice. (**A**) Anti-neuropeptide Y (NPY), anti-parvalbumin (PV) and anti-somatostatin (SST) immunochemistry allow the visualization of specific populations of interneurons within the *stratum oriens* (SO), *stratum pyramidale* (SP), *stratum radiatum* (SR) and *stratum lacunosum moleculare* (SLM) of the CA1 area of hippocampus. (**B**) The density of NPY positive cells was significantly increased in the SR and SLM of Dp(16)1Yey mice. (**C**) The density of PV positive neurons was significantly decreased in the SO of Dp(16)1Yey mice, but remained comparable to WT in the SP where the majority of the PV positive cells are found. (**D**) The density of SST positive neurons was not significantly different between Dp(16)1Yey and WT samples in all four layers of the CA1. Values are expressed as mean ±standard error of the mean. Statistical significance was assessed using Mann-Whitney U-test (**B–D**) with significance set at (*) p<0.05 and (**) p<0.01. Scale bars in (**A**) correspond to 50 μm.

DOI: https://doi.org/10.7554/eLife.31543.016

The following source data is available for figure 5:

**Source data 1.** Quantification of interneuron populations density - full data set.
DOI: https://doi.org/10.7554/eLife.31543.017

neurons was significantly increased in the SP and SLM layers from Dp16 samples (*Figure 5B*). However, no significant differences were observed in the SO and SR layers. PV positive interneurons were predominantly located in the SP layer, with smaller populations present in the SO and SR, but nearly absent in the SLM (*Figure 5A*). The density of PV positive cells was slightly yet significantly decreased in the SO layer of the DS mice (*Figure 5C*). In the SP of Dp16 mice PV cell density trended higher, but did not reach the level of significance (Mann-Whitney U-test, p=0.164). In the SR and SLM layers the PV positive population was similarly very low in both groups. SST positive interneurons were mostly found in the SO and to a lesser extent in the SP, but were nearly absent in the SR and SLM layers (*Figure 5A*). The density of SST positive cells was not significantly different between WT and Dp16 samples in all four layers of CA1 (*Figure 5D*). Our investigation of three markers of inhibitory interneurons revealed subtype and layer specific changes in the CA1 of Dp(16)1Yey mice. Whereas the main populations of PV and SST positive cells in the pyramidal cells layer and stratum oriens respectively were not significantly affected, a significant increase in NPY positive cell density was seen in the SP and SLM layers.

## Discussion

Here we provide the first detailed analysis of *in vivo* CA1 function in the Dp(16)1Yey DS model. During exploration, CA1 pyramidal neurons in these mice showed a significant decrease in ability to burst and produce complex spikes, and while their phase locking to theta oscillations was conserved, spatial encoding and information content was significantly lower. A similar deficit in bursting was observed during post-exploratory rest and was associated at the network level with significant changes in ripple properties, suggesting a deficit in network recruitment and synchronization. These changes were accompanied by an increase in NPY expressing interneurons in the *stratum pyramidale* (SP) and at the *stratum lacunosum moleculare – stratum radiatum* (SLM-SR) interface.

The Dp(16)1Yey model for DS reproduces learning and memory deficits that resemble those seen in people with DS. Behavioral screening and *in vitro* electrophysiology approaches have identified spatial learning and memory deficits associated with a clear long-term potentiation decrease at the Schaffer collateral-CA1 synapses (*Yu et al., 2010a*; *Yu et al., 2010b*). Pharmacological or genetic abolition of NMDA-based plasticity results in more diffuse place fields and more variability in firing both within and across exploration sessions (*Kentros et al., 1998*; *McHugh et al., 1996*; *Cabral et al., 2014*). Here we found that place cells of Dp(16)1Yey mice have larger, more diffuse place fields that could be a consequence of abnormal synaptic plasticity, not solely in the CA1, but potentially also in the upstream trisynaptic circuit, as observed in the Tc1 mice (*Witton et al., 2015*). Although LTP during exploratory behavior is important for the memory reactivation during ripple events, none of the changes we found in ripple characteristics have been observed in models of chemically induced LTP blockade (*Dupret et al., 2010*). These changes, as well as the deficit in bursting and complex spiking, are thus likely to have a different origin. The fact that pyramidal cells recruitment during ripple events is deficient suggests that memory consolidation may be poorer in Dp16 mice and is potentially a key contributor to their learning and memory deficits (*Ramadan et al., 2009*; *Ego-Stengel and Wilson, 2010*).

Hippocampal interneurons can be categorized by the location of their cell bodies and the expression of specific molecular markers (*Somogyi and Klausberger, 2005*). Basket cells, the dominant PV +/NPY-/SST- fast-spiking class of interneurons, play an important role in maintaining network oscillations and synchronizing pyramidal cells to these rhythms (*Cutsuridis and Taxidis, 2013*; *Bartley et al., 2015*; *Schlingloff et al., 2014*; *Royer et al., 2012*). Although an increase in PV expressing cells has been reported in juvenile DS mice (*Chakrabarti et al., 2010*), our results, as well as data from other groups (*Hernández-González et al., 2015*), do not support this observation in adults. Moreover, we did not see physiological differences between Dp(16)1Yey and control mice that would be predicted by changes in basket cells: gamma and theta power, the frequency of ripple events and the phase locking of pyramidal cells to theta are conserved in the Dp16 model. The decrease in gamma power observed *in vitro* under tetanic stimulation in the Ts65Dn model (*Hanson et al., 2013*) was also absent in the Dp(16)1Yey mice under *in vivo* physiological conditions.

The main SST expressing interneurons in the hippocampus are *oriens-lacunosum moleculare* (OLM) cells, defined by the location of their soma in the *stratum oriens* (*Somogyi and Klausberger, 2005*). They also express PV weakly and are likely to be NPY positive (*Somogyi and Klausberger, 2005*; *Milstein et al., 2015*). These neurons regulate pyramidal cells' excitability by directly targeting their dendritic tufts, affecting pyramidal cells bursts and, unlike basket cells, have no effect on the spike timing (*Pangalos et al., 2013*; *Royer et al., 2012*). While we did not find a significant alteration in the size of the OLM interneuron population, functional changes at the cellular level in these neurons could contribute to the drastic decrease in bursts from pyramidal cells in Dp(16)1Yey mice. Further work is required to investigate whether the intrinsic properties of OLM cells are affected in this model.

NPY staining reveals different populations of interneurons: OLM cells in the *stratum oriens* (see above), bistratified cells (BS, also PV positive) and Ivy cells (PV negative) in the *stratum pyramidale* and a newly identified class of SLM-SR interneurons (*Somogyi and Klausberger, 2005*; *Fuentealba et al., 2008*; *Milstein et al., 2015*). Here we found an increase in NPY positive interneurons in the *stratum pyramidale*, putative BS and Ivy cells. These two classes of interneurons possess an overlapping role in the regulation of CA1 pyramidal cells, by targeting processes in the stratum radiatum and oriens (*Cutsuridis and Taxidis, 2013*; *Fuentealba et al., 2008*). Both BS and Ivy interneurons are able to directly control pyramidal cells' excitability by affecting the processing of input

from the CA3 (*Lovett-Barron et al., 2012*; *Fuentealba et al., 2008*). Though it is not clear if Ivy cells are able to induce a similar effect, BS cells are known to control the bursting of pyramidal cells and to switch their firing between single spikes and bursts (*Lovett-Barron et al., 2012*; Cutsuridis and Taxidis, 2012). Dp(16)1Yey mice have an increased density of these NPY positive interneuron and display a strong reduction in their ability to produce bursts of spikes during the two critical phases of spatial learning: exploration and memory consolidation. The shift observed in the biphasic distribution of single versus burst spikes may be caused by an enhanced inhibition from BS and/or Ivy cells. In addition, the increased population of NPY positive interneurons at the SLM-SR border corresponds to a recently identified class of interneurons with unique characteristics (*Milstein et al., 2015*). A similar increase has been reported in the Ts65Dn model (*Hernández-González et al., 2015*). Unlike PV and SST expressing interneurons, these NPY positive cells are able to integrate inputs from both the CA3 and the entorhinal cortex to trigger complex spikes in pyramidal cells (*Milstein et al., 2015*). Enhanced inhibition resulting from increased NPY +interneurons density is thus likely to impact pyramidal cells in the Dp(16)1Yey model and contribute to the decrease in complex spiking seen in these mice.

Deficits in hippocampal plasticity and hippocampal dependent learning and memory are consistent phenotypes observed not only in this model, but also in the other three main mouse models for DS: Ts1Cje, Ts65Dn and Tc1 (*Reeves et al., 1995*; *Sago et al., 1998*; *Siarey et al., 1997*; *Siarey et al., 2005*; *Belichenko et al., 2007*; *O'Doherty et al., 2005*). Over the past decade the extensive study of these DS mice, especially the Ts65Dn model, has led to a main hypothesis that the leading cause for brain phenotypes in DS is over-inhibition by the GABAergic system (*Contestabile et al., 2017*). This theory is originally based on histological and electrophysiological observations showing increases in interneurons, changes in the repartition of inhibitory synapses, and increased GABA induced inhibitory currents (*Chakrabarti et al., 2010*; *Hernández-González et al., 2015*; *Belichenko et al., 2004*; *Belichenko et al., 2009*; *Best et al., 2007*). It is also supported by pharmacological approaches showing a rescue of the learning, memory and LTP deficits using drugs inducing a decrease in inhibitory signaling (*Fernandez et al., 2007*; *Braudeau et al., 2011*; *Martínez-Cué et al., 2013*; *Kleschevnikov et al., 2012*; *Deidda et al., 2015*). As no drug is available to date for treating Down syndrome phenotypes, the GABAergic hypothesis is a leading target for recent clinical trials using broad GABA antagonist compounds.

Thus far, the imbalance between inhibition and excitation and its links to DS-related cognitive deficits has been examined in rodent behavioral and *in vitro* studies, thus more work is required to understand how circuits and microcircuits are affected in the Down syndrome brain. Our work provides the first extensive characterization of the *in vivo* neuronal activity in freely behaving animals and can serve as the basis for the design of novel biomarkers to address the GABAergic theory *in vivo*. Our data indicate that changes occurring in a limited class of interneurons could support abnormal neuronal function at the circuit level, suggesting that drugs targeted to these neurons could prove more successful in treating Down syndrome phenotypes than non-specific GABA antagonists. For technical and practical reasons we did not combine our recordings with drugs in the current study, nonetheless we believe our work could serve as a starting point to screen compounds, together with the behavioral and *in vitro* approaches that have been used successfully thus far. Further studies will be required to investigate potential cause-consequence relationships between changes in interneuron populations and the deficits we observed in pyramidal cell properties.

Though the studies discussed above concur that deficits in DS mice are driven by an enhanced GABAergic inhibition, the exact origin of this excitation/inhibition imbalance remains unclear. Our data supports previous reports suggesting that the pool of specific inhibitory neurons is increased in the hippocampus of DS animals (*Hernández-González et al., 2015*). Very few genes within the trisomic region of Dp(16)1Yey mice are known to have a direct impact on interneuron differentiation. The main candidate mechanisms so far are a decreased response to SHH signaling, most likely though the overexpression of *App* (*Roper et al., 2006*; *Trazzi et al., 2011*; *Xu et al., 2010*) and the direct regulation of embryonic neuronal differentiation by *Olig1* and *Olig2* in the medial ganglionic eminence (MGE; *Chakrabarti et al., 2010*). Interestingly, although the changes observed by these authors in early post-natal hippocampus are not reproduced at adult stages (*Hernández-González et al., 2015*), present study), an abnormal MGE neuron differentiation mechanism could be responsible for the NPY positive populations increase in the Dp(16)1Yey mice, as these cells derive from this embryonic structure (*Tricoire et al., 2010*). The serine-threonine kinase *Dyrk1a* is

also among the top candidates for its ability to affect the neuronal precursors cell cycle, directly or through a synergistic effect with the regulator of calcineurin *Rcan1* (*Arron et al., 2006*; *Hämmerle et al., 2011*). Future work using complex genetic rescue approaches, as developed by Chakrabarti and colleagues (*Chakrabarti et al., 2010*), will help understand the genotype-pheno-type relationships underlying the cellular and electrophysiological abnormalities reported in the present work.

In conclusion, we identified a deficit in the ability of pyramidal cells to generate bursts and com-plex spikes, as well as to synchronize during population events in the Dp(16)1Yey DS model. These changes affect hippocampal function both during exploration and post-exploratory rest periods, i.e. during memory encoding and consolidation phases. The concurrent observation of an increase in NPY expressing interneurons suggests that a subtype specific 'over-suppression' in the DS brain is likely contributing to these changes in physiology. These phenotypes could have a direct role in the spatial learning deficiency observed in these mice and suggest that a targeting of specific neuron subtypes may help recover some of their DS-like impairments.

# Materials and methods

## Key resources table

| Reagent type (species) or resource | Designation | Source or reference | Identifiers | Additional information |
|---|---|---|---|---|
| strain, strain background (mouse, males) | 'Dp(16)1Yey'; 'Dp16' | Jackson Laboratory | 'Stock number: 013530'; 'RRID:MGI:5690055' | |
| antibody | anti-PCP4 rabbit polyclonal IgG | Santa-Cruz | 'sc-74816'; 'RRID:AB_2236566' | 1:200 dilution; frozen sections |
| antibody | anti-NPY | Cell Signaling Technology | '#11976'; 'RRID:AB_2716286' | 1:400 dilution; frozen sections |
| antibody | anti-SST-14 | Peninsula Laboratories | 'T-4103'; 'RRID:AB_518614' | 1:500 dilution; frozen sections |
| antibody | anti-PV | Calbiochem | 'PC255L'; 'RRID:AB_2173906' | 1:1000 dilution; frozen sections |
| antibody | biotinylated goat anti-rabbit IgG antibody | Vector Laboratories | 'BA-1000'; 'RRID:AB_2313606' | |
| commercial assay or kit | RetrievagenA | BD Biosciences | #550524 | antigen retrieval reagent |
| commercial assay or kit | Avidin/Biotin blocking kit | Vector Laboratories | SP-2001 | blocking reagent |
| commercial assay or kit | VECTASTAIN Elite ABC kit | Vector Laboratories | PK-6100 | signal amplification kit |
| commercial assay or kit | NovaRed substrate kit | Vector Laboratories | SK-4800 | revelation kit |
| chemical compound, drug | 'Avertin '; '2, 2, 2-tribromoethanol' | Sigma-Aldrich | T48402 | anaesthetic |

## Subjects

All procedures were approved by the RIKEN Animal Care and Use Committee (project approval numbers H29-2-218(2) and # H29-2-224(3)). The Dp(16)1Yey mouse line was obtained from Jackson Laboratory (www.jax.org, Stock number 013530; RRID:MGI:5690055) and maintained by crossing carrier males with C57BL/6J females. Six Dp(16)1Yey and five WT littermate male mice aged 4 months were used in this study. All mice were group housed by 2 to 5 in ventilated racks with a 12 hr light/dark cycle and *ad libitum* access to food and water. They were single housed after stereo-taxic surgical implantation of microdrives.

## Surgery, recording and histology

Animals were anesthetized using Avertin (2, 2, 2-tribromoethanol; Sigma-Aldrich, 476 mg/kg, i.p.) and implanted with a custom microdrives (manufactured with the assistance of the Advanced Manufacturing Support Team, RIKEN Center for Advanced Photonics, Japan) targeting the dorsal hippocampus (1.6 mm posterior and 1.2 mm right-lateral coordinates from bregma). Microdrives consisted of eight independently adjustable nichrome tetrodes (14 μm) arranged in two rows of 4 and gold plated to reach an impedance of 200 to 250 kΩ. Stainless steel screws placed on the cere-bellum were used as ground and two extra tetrodes placed in the corpus callosum were used as references. Tetrodes were then slowly lowered over the course of several days to reach CA1 stratum

pyramidale, identified by the presence of sharp wave ripples and large amplitude spikes. During this adjustment period, mice were kept in a small circular sleep/rest box (15 cm diameter). Tetrodes were then finely adjusted daily to maximize cell yield before recording. Recordings consisted of 10 laps of exploration on a linear track (170 × 10 cm with 15 cm high plastic walls) bracketed by 30 min pre- and post-exploratory rest/sleep sessions in the familiar circular rest box. Mice were trained for three consecutive days and data recorded on the third day were used for all analyses. Data were acquired using a 32-channel Digital Lynx4S system using Cheetah v5.6.0 acquisition software (Neuralynx). Signals were sampled at 32556 Hz and spike waveforms filtered between 0.6–6 kHz. Position and head direction were concurrently tracked using a pair of red/green light emitting diodes affixed to the microdrive.

At the end of the third recording session, mice were given a lethal dose of Avertin and tetrodes position was marked by electrolytic lesion (50 µA input for ~8 s to each tetrode individually). After transcardial perfusion with 0.9% NaCL/5 mM EGTA followed by 4% paraformaldehyde, brains were collected, post-fixed for 48 hr and embedded in 15% sucrose – 50% OCT. Frozen coronal sections (30 µm) were prepared and labeled by immunohistochemistry using an antibody targeting PCP4, a marker for CA2, to determine the borders of the CA1 area. Sections were incubated in RetrievagenA (#550524, BD Biosciences) and blocked using Avidin/Biotin blocking kit (SP-2001, Vector Laboratories). Sections were then incubated with anti-PCP4 rabbit polyclonal IgG (1:200, Santa-Cruz sc-74816, Santa-Cruz; RRID:AB_2236566) and revealed using biotinylated goat anti-rabbit IgG antibody (1:250, Vector BA-1000, Vector Laboratories; RRID:AB_2313606) followed by 'VECTASTAIN Elite ABC' (PK-6100, Vector Laboratories) and NovaRed substrate kits (SK-4800, Vector Laboratories). Images revealing the positions of the electrodes were acquired using a BZ-X710 light microscope (Keyence).

## Data processing and analyses

### Data processing and unit isolation

Data files from each dataset were split by manually recorded trial timestamps using EventSessionSplitter software (Neuralynx). Artifacts in the animal's positional values caused by the obscuring of diodes were removed using a custom written algorithm and the positional data was smoothed with a Gaussian kernel of 0.05 standard deviation (SD) width. Single units were isolated manually, in SpikeSort3D software (Neuralynx), by drawing cluster boundaries around the 3D projection of the spike features. The boundaries were tracked across same-day recording trials to ensure cluster stability. Clusters that had greater than 0.5% of their spikes violate a minimum 2 ms inter-spike interval (ISI), fire less than 50 spikes or display an isolation distance measure (Schmitzer-Torbert et al., 2005) <10 were excluded from further analyses. Remaining units were classified as pyramidal cells if their average spike width was >200 µS and had a complex spike index ≥5, (CSI; McHugh et al., 1996). Animal velocity was calculated based on recorded position values and corresponding timestamps and then smoothed with a 2.5 SD Gaussian kernel. All subsequent analyses were performed in MAT-LAB (MathWorks), using custom written scripts.

### Single unit and place field properties

Firing rate maps were calculated by dividing the number of spikes falling into each 1 cm x 1 cm spatial bin by the total occupancy time of that bin and were subsequently smoothed with a 1 SD Gaussian kernel; unvisited bins and time periods when animal's velocity was below 2 cm/sec were excluded. For a subset of analyses such as 'directionality index' and 'population vectors' a 'firing rate curve' i.e. a directionally-sensitive 1D representation of a 'firing rate map' was used. For the 'firing rate curve' calculation we first detect 'laps' – time periods when mouse was running along the track by using custom detection algorithm (Polygalov, 2017). The quality of lap detection was controlled visually. The left firing curve was defined then as total number of spikes fired across all left laps within each spatial bin divided by total time mouse spent in that spatial bin. The right firing curve was calculated similarly. Peak firing rate was defined as the rate in the spatial bin containing the maximal firing rate value within each rate map. Mean firing rate was calculated by dividing the number of spikes which occurred within periods when velocity exceed 2 cm/sec by that period's duration and followed averaging of these values. A place field was defined as a set of contiguous spatial bins surrounding the bin where the maximal firing rate was observed. In-field mean firing rate

was the total number of spikes emitted by the cell while the mouse was in the place field with the highest peak firing rate (main place field of that cell) divided by the total time spent by the mouse in this field. Out-field mean firing rate was the number of spikes emitted by the cell in all spatial bins outside of the main field of that cell divided by the total time spent by the mouse outside of the main place field. Place cells were required to have a minimum field size equivalent to six bins (1 cm x 1 cm bin size), a mean firing rate >0.2 Hz, a peak firing rate >1.0 Hz and a positive signal to noise ratio (SNR, *Resnik et al., 2012*). Place field size was defined as the number of spatial bins where place cell field firing exceeded 20% of the peak firing rate. Rate map sparsity was computed as previously described (*Resnik et al., 2012*). Firing rate map 'sparsity' is a number ranging from 0 to 1, were 0 correspond to a firing rate map which consists of equal firing rate values in every visited spatial bin. Firing rate map with sparsity value one corresponds to the case when all the spikes generated by any given cell were fit in a single spatial bin. The directionality index (DI) was calculated for the subset of place cells with their main field located in the middle 80% of the track. DI was defined as the absolute value of ((FR_lm - FR_rm)/(FR_lm + FR_rm)) where FR_lm is the mean firing rate across all left laps of a given trial and FR_rm is the mean firing rate across all right laps of a given trial. Spatial Information (bits per second) was calculated as previously reported (*Skaggs et al., 1992*).

## Complex spike index and burst analysis

The Complex Spike Index (CSI) is defined as CSI = 100 * (pos - neg), where 'pos' is the number of inter-spike intervals positively contributing to the CSI, that is, preceding spikes with larger amplitudes and following spikes with smaller amplitudes (complex bursts) occurring within 3 ms (refractory period) and 15 ms (maximum inter-spike interval defining a burst); 'neg' is the number of inter-spike intervals that contribute negatively to CSI, i.e. violating either or both these rules. A burst was defined as least two spikes occurring within a 10 ms time bin and all burst detection and analyses were performed using Matlab code previously described (*Bakkum et al., 2013*).

## Population vector analysis

Population vector (PV) was defined as a set of firing rate values generated by all place cells from all mice within each group and corresponding to each particular spatial bin of the linear track. Left and right laps were treated separately. Note that for this analysis all place cells (including cells with place field at the ends of the track) were used. For auto- and cross-correlation of PVs a Spearman correlation coefficient was used.

## Local field potential (LFP) analysis

The raw LFP data were downsampled using custom software written in C to 1627.8 Hz (a factor of 20), followed by quality control measure and channel selection via visual inspection. A low-pass filter with a cut-off frequency equal to half the target sampling frequency was applied to the LFP prior to downsampling to prevent signal distortion.

*Power Spectral Density (PSD)* during exploratory behavior was calculated by using Welch's averaged modified periodogram method (*pwelch* function in Matlab) with 2048 samples window size (1.26 s), 50% overlap and 4096 FFT points (2.52 s) resulting a time-varying spectrogram. A PSD curves corresponding to time bins when animal's velocity was above 2 cm/sec were averaged yielding single PSD curve for each behavioral trial. In order to account for power fluctuations caused by difference in position/impedance of the electrodes and make PSD values comparable across mice we normalize each PSD curve by its own mean power within delta (1–3 Hz) band.

## Ripple detection

Ripple events were detected using modifications to methods described previously (*Csicsvari et al., 1999*). Wide band LFP were band-pass filtered between 80 and 250 Hz using 69 orders Kaiser-window FIR zero-phase shift filter. The absolute value of Hilbert transform was then smoothed with 50 ms Gaussian window and candidate ripple events were detected as periods where magnitude exceeded 3SD above the mean for >30 ms. Of the events, the initiation and termination periods were defined as periods when the magnitude returned to the mean. Summed multi-unit activity (MUA) across all neurons was converted to instantaneous firing rate and smoothed, to allow

detection of firing bursts using the same thresholds as described for LFP. Candidate ripple events not coincident with MUA bursts were excluded from subsequent analysis. For cleaner detection of ripple frequency for each ripple, a multitaper method was performed on the product of each filtered ripple waveform and a Hanning window of the same length.

To control for the impact of the significant decrease in normal ripple amplitude observed in the Dp16 mice on the quantification of ripple-related pyramidal cell spiking ripple we adjusted the threshold and repeated the ripple detection algorithm detection. At both a more permissive (2 SD) and a more restrictive (6 SD) threshold, the changes in spiking were observed (see *Figure 4—figure supplement 2*).

## Theta modulation of pyramidal cells

The phase relationship between spikes and theta LFP was calculated as previously described (*Siapas et al., 2005*). Briefly, LFP traces were band-pass filtered in the theta band (6–12 Hz). Instantaneous theta phase was derived from the Hilbert-transformed theta filtered signal. Peaks and troughs were assigned 0 and 180 degree phases respectively, with spike phase calculated using interpolation, a method not sensitive to theta wave asymmetry. The resultant phases were converted to firing probability histograms (10 degree bin size), only when velocity exceeded 6 cm/sec. Significance of the phase locking, preferred firing phase, strength of modulation and statistical comparison of phase values were calculated using functions from Circular Statistics Toolbox (*Berens, 2009*).

## Immunohistochemistry for interneuron quantification

Brains from adult mice (3.5 to 4 months old) were collected and fixed in 4% PFA. Equivalent frozen sections (30 µm thickness) from Dp(16)1Yey (N = 4) and their WT littermates (N = 4) were selected according to common landmarks at positions equivalent to the regions targeted in the *in vivo* electrophysiology recording experiments (*Paxinos and Franklin, 2001*). Sections were incubated in citrate-EDTA buffer (10 mM citrate, 1 mM EDTA; 80°C for 20 min) and blocked using Avidin/Biotin blocking kit (SP-2001, Vector Laboratories). Sections were then incubated with anti-NPY (1:400, #11976, Cell Signaling Technology; RRID:AB_2716286), anti-SST-14 (1:500, T-4103, Peninsula Laboratories; RRID:AB_518614) or anti-PV (1:1 000, PC255L, Calbiochem; RRID:AB_2173906) and revealed using biotinylated goat anti-rabbit IgG antibody (1:250, Vector BA-1000, Vector Laboratories; RRID:AB_2313606) followed by 'VECTASTAIN Elite ABC' (PK-6100, Vector Laboratories) and NovaRed substrate kits (SK-4800, Vector Laboratories). Images were acquired using a BZ-X710 light microscope (Keyence) and processed using NIH ImageJ software for particle counting and area calculation. A total of six hippocampal images were counted for each animal in a blinded manner.

## Acknowledgements

We are grateful to RIKEN Brain Science Institute (BSI) Research Resource Center for the support with mice maintenance and Dr. Kenji Yamazawa and the Advanced Manufacturing Support Team, RIKEN Center for Advanced Photonics for assistance in microdrive production. We also thank all the members of the laboratory for Neurogenetics and the laboratory for Circuit and Behavioral Physiology for helpful comments and advice. This work was supported by RIKEN-BSI [to KY and TJM]. No external funding was received for this work.

## Additional information

### Funding

| Funder | Author |
| --- | --- |
| RIKEN Brain Science Institute | Kazuhiro Yamakawa<br>Thomas J McHugh |

The funders had no role in study design, data collection and interpretation, or the decision to submit the work for publication.

## Author contributions
Matthieu Raveau, Conceptualization, Formal analysis, Investigation, Visualization, Methodology, Writing—original draft, Writing—review and editing; Denis Polygalov, Conceptualization, Data curation, Software, Formal analysis, Validation, Visualization, Methodology, Writing—original draft, Writing—review and editing; Roman Boehringer, Resources, Visualization, Methodology, Writing—original draft; Kenji Amano, Resources, Writing—review and editing; Kazuhiro Yamakawa, Conceptualization, Supervision, Funding acquisition, Writing—review and editing; Thomas J McHugh, Resources, Software, Formal analysis, Supervision, Funding acquisition, Visualization, Writing—original draft, Project administration, Writing—review and editing

## Author ORCIDs
Matthieu Raveau ⓘ http://orcid.org/0000-0002-3951-3861
Denis Polygalov ⓘ https://orcid.org/0000-0002-8165-5257
Roman Boehringer ⓘ https://orcid.org/0000-0003-2856-3262
Thomas J McHugh ⓘ http://orcid.org/0000-0002-1243-5189

## Ethics
Animal experimentation: All handling and experiments were conducted in accordance with the protocols approved by the RIKEN Animal Care and Use Committee (#H29-2-218(2), # H29-2-224(3)).

## Decision letter and Author response
Decision letter https://doi.org/10.7554/eLife.31543.022
Author response https://doi.org/10.7554/eLife.31543.023

## Additional files

### Supplementary files
• Transparent reporting form
DOI: https://doi.org/10.7554/eLife.31543.018

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
