## [Decision Letter]

[Editors’ note: this article was originally rejected after discussions between the reviewers, but the authors were invited to resubmit after an appeal against the decision.]

Thank you for submitting your work entitled "Alterations of *in vivo* CA1 network activity in Dp(16)1Yey Down syndrome model mice" for consideration by *eLife*. Your article has been reviewed by two peer reviewers, and the evaluation has been overseen by a Reviewing Editor and a Senior Editor. The reviewers have opted to remain anonymous.

Our decision has been reached after consultation between the reviewers. Based on these discussions and the individual reviews below, we regret to inform you that your work will not be considered further for publication in *eLife*.

The reviewers seem to find that this study is interesting in that it reports altered network activities in a mouse model of Down syndrome, but the results are largely descriptive and do not provide sufficient insights into whether and how the decreased network activity in Dp(16)1Yey Down syndrome model mice affects hippocampal memory in these animals.

*Reviewer #1:*

This manuscript addresses the question of what circuit alterations in a mouse model of Down syndrome, the Dp(16)1Yey model, give rise to learning and memory deficits. The authors recorded from hippocampal CA1 place cells while animals ran on a linear track and while they rested. They found that place cells in the Dp mice have wider place fields with less prospective coding and local field potentials (LFPs) in the Dp mice show less ripple oscillations, both of which are linked to reduced bursty firing of CA1 cells. Together with the finding of more NPY-positive neurons in certain layers of CA1, the authors conclude that memory encoding and consolidation are impaired due to altered bursty firing caused by over inhibition. The manuscript is well written. Although I wish the story could be more exciting or specific, I believe this study provides a more comprehensive account of *in vivo* neural circuit deficits in a Down syndrome model than previous literature and is valuable to the field of neurological diseases.

1) The authors used lots of bar plots to present data. However, many of the measures, for example firing rate, spatial information, and spike number per SWR, are not necessarily normally distributed. Mean and standard error (s.e.) values may not correctly describe the data samples. I believe plotting the whole distribution, using either cumulative or histogram, is a better way of showing the data. Second, the Mann-Whitney U-test used in the manuscript does not compare mean values, but more appropriately for median values. It is inappropriate to place significance symbols (like ***) over two bars, which show mean values.

2) Related to the concern above, the inter-spike interval (ISI) mean values in Figure 3 clearly do not describe the distributions in Figure 3. More importantly, it is unclear how the mean and s.e. values were computed. Why the mean values in Figure 3 are so small (4 – 8 ms), while it is clear that vast majority of ISIs were larger than 10ms? This discrepancy needs to be resolved.

*Reviewer #2:*

Raveau and colleagues perform extracellular recordings from the CA1 hippocampal region of behaving Down Syndrome (Dp16) model mice, alongside controls, and describe the observed differences. There is less spatial specificity, bidirectional coding, and individual and population bursting in Dp16 pyramidal cells. These animals also display more neuropeptide Y-positive cells in the CA1 pyramidal cell layer. The analysis is mostly appropriate though I had some addressable concerns in a couple of areas. The writing was fine except for some minor redundancies and problematic reference citations. With minor revision, this work will be helpful for researchers working to understanding circuit and cognitive deficiencies in Down Syndrome, though I don't believe the level of insights provided are of high general impact.

I have some concern regarding the various analyses.

1) For the ISI distribution analysis, I am concerned that a few outlier neurons may be contributing disproportionately to the distributions. To account for this possibility, please show the distributions of average ISIs for the neurons underlying Figure 3.

2) In the ripple event/firing rate analysis, I am concerned that the lower amplitude of events in Dp16 mice may make detection of their onsets and offsets less reliable, thus confounding the other measures (Figure 4). I noted that the onset and offsets of MUA bursts were also recorded. Can you please also provide these measures for the concurrent MUA bursts (i.e. to provide spikes per MUA burst, participation% , etc.) to make sure that these differences are robust?

[Editors’ note: what now follows is the decision letter after the authors submitted for further consideration.]

Thank you for resubmitting your article "Alterations of *in vivo* CA1 network activity in Dp(16)1Yey Down syndrome model mice" for consideration by *eLife*. Your article has been reviewed by two peer reviewers, and the evaluation has been overseen by a Reviewing Editor and a Senior Editor. The reviewers have opted to remain anonymous.

The reviewers have discussed the reviews with one another and the Reviewing Editor has drafted this decision to help you prepare a revised submission.

The reviewers find that the approach is novel and the data are interesting and solid. However, the manuscript still contains a point not addressed, and, more importantly, the reviewers are seriously concerned about the possibility that some of key conclusions may not be associated with the pathology of Down syndrome. Therefore, we request that you fully address these points by performing additional analyses and carefully discuss all the arguments in the main text in a balanced manner.

*Reviewer #1:*

This revised manuscript describes *in vivo* neurophysiological deficits in the hippocampus of a mouse model of Down syndrome. The main finding is that reduced bursty firing of hippocampal place cells leads to less precise place fields during running, suggesting a deficit in memory encoding, and less participation of cells in sharp-wave ripples during resting, suggesting a deficit in memory consolidation. The authors further performed immunohistochemistry to suggest that the reduction in bursty firing is related to changes in the number of NPY-positive interneurons, which results in over-inhibition in the mouse model. The revision has improved the writing and data presentation. The authors have also addressed the potential impacts of their finding on the understanding and treatment of Down syndrome. The quality of the manuscript has been improved. Also, I believe the issue of general impact has been strengthened to a degree that I believe the paper should be published.

*Reviewer #2:*

My assessment remains that the different observations made in this manuscript of differences between Dp16 and wild-type mice (e.g. spatial firing specificity, bidirectional coding, bursting propensity, spatial distribution of interneuron subclasses), while of interest for a better understanding of mouse models of Down Syndrome, do not aggregate to important mechanistic insights that will be of high general impact. That these changes stem from increased NPY expression in the pyramidal and LM layers is still speculative at this stage.

One other point regarding the ripple/MUA burst analysis, I think the authors missed my suggestion to perform the analysis on MUA bursts that are *concurrent* with ripples, to avoid the issue they mentioned of contamination from non-ripple periods. I also think the outcomes of this analysis should be presented in the main text, not just for reviewers.

---

## [Author Response]

[Editors’ note: the author responses to the first round of peer review follow.]

Reviewer #1:

*This manuscript addresses the question of what circuit alterations in a mouse model of Down syndrome, the Dp(16)1Yey model, give rise to learning and memory deficits. The authors recorded from hippocampal CA1 place cells while animals ran on a linear track and while they rested. They found that place cells in the Dp mice have wider place fields with less prospective coding and local field potentials (LFPs) in the Dp mice show less ripple oscillations, both of which are linked to reduced bursty firing of CA1 cells. Together with the finding of more NPY-positive neurons in certain layers of CA1, the authors conclude that memory encoding and consolidation are impaired due to altered bursty firing caused by over inhibition. The manuscript is well written. Although I wish the story could be more exciting or specific, I believe this study provides a more comprehensive account of* in vivo *neural circuit deficits in a Down syndrome model than previous literature and is valuable to the field of neurological diseases.*

We thank the reviewer for the helpful and supportive comments. While we agree that the characterization of the physiological changes in a disease model mouse can result in a largely descriptive study, we would like to emphasize to both you and the readers why we feel this work is important. One reason it has been difficult to generate simple and targetable hypotheses concerning the underlying mechanisms of cognitive deficits in DS patients is the lack of data on circuit level functional changes. To make this clear, and more explicitly position our data in light of current literature and hypotheses, we have added a paragraph to the Discussion on this issue. Specifically, we feel our study supports and extends the “GABAergic hypothesis” of DS cognitive dysfunction and can serve as the basis for the design of novel biomarkers to address and test this theory *in vivo*.

1) The authors used lots of bar plots to present data. However, many of the measures, for example firing rate, spatial information, and spike number per SWR, are not necessarily normally distributed. Mean and standard error (s.e.) values may not correctly describe the data samples. I believe plotting the whole distribution, using either cumulative or histogram, is a better way of showing the data. Second, the Mann-Whitney U-test used in the manuscript does not compare mean values, but more appropriately for median values. It is inappropriate to place significance symbols (like ***) over two bars, which show mean values.

We thank the reviewer for this excellent suggestion and completely agree. We have now revised all the figures that represented large sets of data for which we used the Mann-Whitney U-test to assess the statistical significance.

Figure 1 was modified to describe the distribution of the whole population of neurons recorded. We opted for a combination of graphs representing cumulative frequencies for all these parameters and added whisker box plots in the lower quadrant of each panel. The results of the significance test were added on top of the box plot graphs for parameters where the statistical significance was reached.

In a similar way, we modified the Figure 3 to include a more detailed representation of the distribution of the values for each parameter (panels A, B, C and E). The former panels D and E were inverted in the present version in order to facilitate the description of the figure in the Results section.

Finally, the Figure 4 was edited to replace the standard histograms on panels G and H by whiskers box plots more appropriate to describe the distribution of the whole population of neurons recorded. Accordingly, the legends to these figures were edited to reflect these changes.

2) Related to the concern above, the inter-spike interval (ISI) mean values in Figure 3 clearly do not describe the distributions in Figure 3. More importantly, it is unclear how the mean and s.e. values were computed. Why the mean values in Figure 3 are so small (4 – 8 ms), while it is clear that vast majority of ISIs were larger than 10ms? This discrepancy needs to be resolved.

We apologize for the confusion caused by our mislabeling Figure 3.

Figure 3 in the original manuscript (now corresponding to Figure 3 in the revised version) represents the distribution of ISI values from all spike trains recorded from CA1 pyramidal cells during the post-run rest. This includes frequent short ISIs (2-15ms) generated during bursts, as well as the highly variable long ISIs representing time between bursts and/or single spikes. The main difference we wished to highlight in this figure is the decrease in the short-latency ‘burst’ peak in the Dp16 mice.

Figure 3 (now Figure 3) was poorly described and labeled in the original manuscript. This graph represents the average of the mode of the ISI values of single pyramidal cells. To generate these data we first identified the most frequent ISI for each individual cell (the mode of the distribution), which typically is a small value due to the regularity and frequency of burst events. Then we averaged these values across the cells of each genotype. A longer average ISI mode without a significant change in the mean firing rate supports our conclusion that spikes generated by CA1 pyramidal cells in Dp16 mice were sparser and less likely to be involved in bursts.

The Figure 3 was edited to improve the representation of the dataset and the label corrected to “Interspike interval mode (ms)”. The manuscript was edited to describe accurately these results in the Results section (subsection “Decreased bursting during memory consolidation in Dp(16)1Yey mice”).

Reviewer #2:Raveau and colleagues perform extracellular recordings from the CA1 hippocampal region of behaving Down Syndrome (Dp16) model mice, alongside controls, and describe the observed differences. There is less spatial specificity, bidirectional coding, and individual and population bursting in Dp16 pyramidal cells. These animals also display more neuropeptide Y-positive cells in the CA1 pyramidal cell layer. The analysis is mostly appropriate though I had some addressable concerns in a couple of areas. The writing was fine except for some minor redundancies and problematic reference citations. With minor revision, this work will be helpful for researchers working to understanding circuit and cognitive deficiencies in Down Syndrome, though I don't believe the level of insights provided are of high general impact.

We are grateful for the reviewers input and support. As we stated in our reply to reviewer #2 above, we agree that it can be challenging to identify a precise mechanism underlying phenotypes observed in disease model mice. However, given the high degree of genetic change and the plethora of developmental and adult phenotypes that accompany DS, it has been difficult to generate simple and targetable hypotheses concerning the underlying mechanisms of cognitive deficits in these patients. We feel that our data can be very useful in extending and refining current thinking, particularly the idea that cognitive phenotypes associated with DS stem from a shift in the I/E balance in the hippocampus. We have modified our Discussion to more directly reflect this and are more explicit in stating how we believe our findings can help more the field forward.

I have some concern regarding the various analyses.1) For the ISI distribution analysis, I am concerned that a few outlier neurons may be contributing disproportionately to the distributions. To account for this possibility, please show the distributions of average ISIs for the neurons underlying Figure 3.

We are happy to address this concern for the reviewer. First, as detailed in response to point #2 of reviewer 1 above, we apologize for the confusion caused by our poor description of Figure 3 in the original manuscript. Figure 3 (now Figure 3 in the revised version) represents the distribution of all ISI values from all spike trains recorded from CA1 pyramidal cells during the post-run rest (not averaged by neuron). This includes the frequent short ISIs (2-15ms) generated during bursts, as well as the highly variable long ISIs representing time between bursts and/or single spikes. The main difference we wished to highlight in this figure is the decrease in the short-latency ‘burst’ peak in the Dp16 mice. To address your concern that this result is impact by a few outlier neurons we did calculate the average ISI for each pyramidal cell and compared these values across groups. Both groups demonstrated similar non-normal population distributions (Author response image 1: Kurtosis: WT: 10.41, Dp16: 7.06; D’Agostino K^2^: WT: 154.4, Dp16: 124.5). Further, there was no significant difference between genotypes, with the group exhibiting similar medians (WT: 2443 ms, Dp16: 2226 ms; Mann Whitney p=0.47) and quartile values (Q1: WT: 1227 ms, Dp16: 982 ms; Q3: WT: 4752 ms, Dp16: 4672 ms).

**Author response image 1. respfig1:** Population distribution of the mean ISI of pyramidal cells in wild-type (blue) and Dp(16) (red) pyramidal cells.

An important point of note is that for a single neuron the distribution of ISIs is large and non-normal, ranging from the short ISIs generated during bursts or high-frequency spiking to seconds long ISIs frequently observed in pyramidal cells during rest. Thus, we feel that the data presented in the original Figure 3 (now Figure 3) which plots the average of the mode of the ISI values of single pyramidal cells is a more informative measure. To generate these data we first identified the most frequent ISI for each individual cell (the mode of the distribution), which typically is a small value due to the regularity and frequency of burst events. Then we averaged these values across the cells of each genotype. A longer average ISI mode without a significant change in the mean firing rate supports our conclusion that spikes generated by CA1 pyramidal cells in Dp16 mice were sparser and less likely to be involved in bursts. We realize this figure was not well described in the previous version of the paper and apologize for the error.

Finally, to demonstrate to the reviewer that this increase in the mode is independent of the mean ISI, we divided pyramidal cells of each group into bins based on their average ISI and calculated the mean ISI mode across these subpopulations. As see in Author response image 2, the increase in the mode is robust across the entire population and most pronounced in the bins below 4000 ms.

**Author response image 2. respfig2:** The average ISI mode is larger in the Dp(16) pyramidal cells across all average ISI bin. (0-2000ms WT n=92, Dp16 n=99; 2000-4000 ms WT n=53, Dp16 n=48; 4000-6000ms WT n=40, Dp16 n=19; 6000-8000 ms WT n=9, Dp16 n=17; 8000-10000ms WT n=7, Dp16 n=9; >10000ms WT n=12, Dp16 n=16)

We have now updated the text in the subsection “Decreased bursting during memory consolidation in Dp(16)1Yey mice” to better explain this analysis and updated the legend of Figure 3.

2) In the ripple event/firing rate analysis, I am concerned that the lower amplitude of events in Dp16 mice may make detection of their onsets and offsets less reliable, thus confounding the other measures (Figure 4). I noted that the onset and offsets of MUA bursts were also recorded. Can you please also provide these measures for the concurrent MUA bursts (i.e. to provide spikes per MUA burst, participation% , etc.) to make sure that these differences are robust?

As the reviewer notes, the average normalized amplitude of ripple events in Dp16 mutant mice is lower than in WT. Nonetheless, we would like to explain why we think that the properties of our ripple detection algorithm allow our spiking analysis to remain robust.

First we would like to clarify our standard approach. The onset and offset times of ripple events were detected on each LFP channel individually, based on how many standard deviations instantaneous LFP power in the ripple frequency band (80-250 Hz) deflect from that channel’s mean LFP power value measured in the same band, using a threshold of 3 s.d., which is standard in the field. The candidate ripple events detected based on the normalized LFP power are then checked for co-occurrence with the underlying MUA bursts. The MUA bursts are detected by converting spike trains generated by multiple putative pyramidal cells into single instantaneous firing rate curve, which is also normalized in the same way as the instantaneous LFP power, yielding start and stop timestamps of the MUA bursts. Only these candidate ripple events detected from the LFP signal which co-occurred with supra-threshold MUA bursts were used then for the further analysis.

Given the normalization and combined LFP/MUA approach we are confident we are reliably detecting true ripple events and not omitting events in the DS mice. Note that we observed no change in the number of events detected with this approach. However we are happy to address the reviewer’s concern as requested. We have now repeated the analysis using three variations of the ripple detection algorithm. First, as asked, we detected candidate “ripple” event using only the significant increase in MUA activity, independent of the LFP recordings, and use the threshold crossing times of the MUA activity to define event onset and offset. While this approach detects significantly more events in both groups of mice, likely due to the inclusion of non-ripple related firing rate changes, the decrease in single unit spiking and the shortening of the ripple duration are still significantly altered in the Dp16 mice. With this method the change in ripple participation, albeit lower in the Dp16 mice, was no longer significant as p=0.08. This could be explained by the 2x increase in event number being contaminated with non-ripple periods. A table of results is included below for the reviewer:

**LFP detection – 3SD****MUA analysis****WT****Dp16****p-value****WT****Dp16****p-value***Spikes per ripple*0.241±0.0090.218±0.010**0.0041**0.204±0.0110.184±0.010**0.0219***Firing rate in ripple*0.603±0.0220.546±0.025**0.0041**0.509±0.0260.461±0.024**0.0219***Participation in ripple (%)*15.807±0.51114.661±0.591**0.0127**12.070±0.48411.593±0.530***0.0834****Nb. Ripple per min*26.400±1.40024.833±3.2700.692156.000±2.28058.833±2.9820.4846*Mean Inter-Ripple Interval (s)*2.198±0.1222.646±0.4070.35801.071±0.0461.036±0.0500.6187*Mean ripple duration (ms)*99.373±1.46989.267±2.155**0.0049**92.231±5.16973.079±4.517**0.0206***Peak PSD frequency (Hz)*143.573±4.228150.758±2.2380.1919*Ripple amplitude (nSD)*1.795±0.0532.041±0.042**0.0067***Nb. spikes within ripples*38.598 ± 5.077 27.096 ± 5.2640.154935.980 ± 5.76124.185 ± 5.2250.1634

To further test the robustness of our observation and increase our confidence we were restricting our analysis to true ripple periods, we repeated the ripple detection procedure in all mice using a lower (2 s.d.) and a higher (6 s.d.) threshold for the LFP ripple power measure, again combined with MUA cooccurrence. Although this dramatically changes the number of events detected, once more we found that ripple related spiking of pyramidal cells in the Dp16 mice was significantly altered. We have included these additional analyses as a new supplemental figure (Figure 4—figure supplement 2) and now refer to it in the subsection “Sharp wave ripples properties are affected and place cells fire less during ripples in Dp(16)1Yey CA1”.

[Editors’ note: the author responses to the re-review follow.]

Reviewer #2:My assessment remains that the different observations made in this manuscript of differences between Dp16 and wild-type mice (e.g. spatial firing specificity, bidirectional coding, bursting propensity, spatial distribution of interneuron subclasses), while of interest for a better understanding of mouse models of Down Syndrome, do not aggregate to important mechanistic insights that will be of high general impact. That these changes stem from increased NPY expression in the pyramidal and LM layers is still speculative at this stage.

As stated in our previous point-by-point response, we fully recognize the challenges in the identification of precise mechanisms underlying the phenotypes observed in disease model mice. This is an issue not just for DS, but extends to other models of disease and cognitive dysfunction as well. Nonetheless, there is clear support and interest in the field for these types of studies, as borne out by the continued publication and citation of work addressing the physiological changes in circuit function in disease model organisms. For several recent examples similar in approach to our current work please see: 22q11.2 model (schizophrenia): Zaremba et al., Nat. Neuro. 2017; MeCP2 -/- (Rett syndrome): Lu et al., Neuron 2016; rTg4510 (AD): Cheng and Ji, *eLife*, 2013; PP2B cKO (schizophrenia): Suh et al., Neuron, 2013; Fmr1-Null (fragile X): Talbot et al., Neuron 2018.

Further, we agree that our results do not allow us to establish a cause-consequence relationship between our electrophysiological and cell count observations. For exactly that reason, in the previous and current versions of the manuscript we do not state that the increase in NPY interneuron populations is directly responsible for the abnormal properties of CA1 pyramidal cells. We frame our conclusions on this matter as a hypothesis and acknowledge further study is required to fully test it. In order to emphasize this point, we added the following sentence to the Discussion:

“Further studies will be required to investigate potential cause-consequence relationships between changes in interneuron populations and the deficits we observed in pyramidal cell properties.”

Finally, regarding the issues of mechanism and impact the reviewer raises, perhaps not surprisingly, we respectfully disagree. We remain confident that our high-quality *in vivo* physiological characterization of a well-established mouse model of DS, along with the anatomical data, lend strong support to the existing hypothesis that cognitive deficits in DS are related to increased inhibition. Most importantly, these novel data can serve to provide a benchmark for circuit function in the brain of DS model mice and prove useful in the design and testing of future therapeutic interventions.

One other point regarding the ripple/MUA burst analysis, I think the authors missed my suggestion to perform the analysis on MUA bursts that are concurrent with ripples, to avoid the issue they mentioned of contamination from non-ripple periods. I also think the outcomes of this analysis should be presented in the main text, not just for reviewers.

We apologies if we misunderstood this suggestion in the previous set of revisions. We believe this refers to your previous major comment #3:

*“3) In the ripple event/firing rate analysis, I am concerned that the lower amplitude of events in Dp16 mice may make detection of their onsets and offsets less reliable, thus confounding the other measures (Figure 4). I noted that the onset and offsets of MUA bursts were also recorded. Can you please also provide these measures for the concurrent MUA bursts (i.e. to provide spikes per MUA burst, participation% , etc.) to make sure that these differences are robust*?”

As requested, we reanalyzed all the parameters regarding ripple events and pyramidal cell activity using the multiunit activity to detect onsets and offsets of ripple events. The results are highly similar to those obtained when we used the LFP to detect event onset and offset (Figure 4). Ripples detected through this method were significantly shorter and of lower amplitude in Dp16 animals, and the firing and ripple participation of individual pyramidal cells were significantly decreased in the Dp16 group. As suggested, we summarized these information in a supplementary figure (Figure 4—figure supplement 3).

Accordingly, the following sentence was added to the Results section:

“To further confirm that these changes were not attributable simple to the smaller amplitude of ripples in the Dp16 group we also detected the onset and offset of the events using only multiunit spiking activity and again observed a significant decrease in ripple amplitude, duration, number of spikes and participation in ripple events (Figure 4—figure supplement 3).”